# Temperature and Crystalline Orientation-Dependent Plastic Deformation of FeNiCrCoMn High-Entropy Alloy by Molecular Dynamics Simulation

**Fuan Yang** [1], **Jun Cai** [1,*], **Yong Zhang** [2] and **Junpin Lin** [2]

1   School of Nuclear Science and Engineering, North China Electric Power University, Beijing 102206, China
2   State Key Laboratory for Advanced Metals and Materials, University of Science and Technology Beijing, Beijing 100083, China
*   Correspondence: caijun@ncepu.edu.cn

**Abstract:** The effect of the crystallographic direction and temperature on the mechanical properties of an FeNiCrCoMn high-entropy alloy (HEA) is explored by molecular dynamics simulations. The calculated static properties are in agreement with the respective experimental/early theoretical results. The calculated compressive yield stress along the <010> direction of a single crystal/polycrystal is the same in order of magnitude as the experimental results. The yield stress and Young's modulus of the single crystal show strong anisotropy. Unlike the single crystal, the polycrystal behaves as an isotropic and has strong ductility. It is found that the dislocations produced in the plastic deformation process of the HEA are mainly 1/6<112> Shockley dislocations. The dislocations produced under normal stress loads are far more than that in the shearing process. FCC transformation into HCP does not occur almost until yield stress appears. The yield stress, yield strain, and Young's modulus reduce gradually with increasing temperature. The modulus of the single/double crystal under compressive and tensile loads presents an obvious asymmetry, while there is only a small difference in the polycrystal. The strain point is found to be the same for stress yielding, FCC-HCP phase transition, and dislocation density, varying from slow to fast with strain at the considered temperature.

**Keywords:** FeNiCrCoMn high-entropy alloy; dislocation density; yield stress; Young's modulus; molecular dynamics

## 1. Introduction

The mechanical properties of high-entropy alloys (HEAs) are closely related to the dislocation motion and HEAs' structure [1–5]. At present, related experimental and theoretical works are being carried out extensively. For example, Laplanche et al. [6] found in their experiments that CoNiCrFeMn undergoes a large number of twin deformations under pressure at low temperatures. The twin boundaries may increase the resistance of dislocation slip. He et al. [7] found, experimentally, that the deformation of an FeCoNi-CrMn HEA under pressure at high temperatures is dominated by the dislocation climbing at strain rates greater than $2 \times 10^{-5}/s$. At such high temperatures, Mn and Cr-enriched deposited phases are formed at lower strain rates and result in a redistribution of the alloy constituents. The formation of the deposited phases consumes the Mn and Cr atoms in the alloy and results in the weakening mechanical strength of the alloy. Tian et al. [8] performed quasi-static tensile experiments on an ultrafine crystalline 1% C-CoCrFeMnNi HEA at 77–823 K. It was found that the deformation mechanism of the material shifts to dislocation and grain boundary slip with increasing temperature, and the strength and ductility of the material decrease. Sun et al. [9] found that the deformation mechanism in forged a CrMnFeCoNi HEA is temperature-dependent. With an increase in temperature, the deformation mechanism changes from mixed dislocation slip and twin deformation to wavy slip and dislocation cell formation.

However, in an experiment, a material's microstructure during its deformation cannot be determined in real-time. Molecular dynamics (MD) is a powerful theoretical tool that can be used to directly observe the microstructure of an HEA during the deformation of the material. Alhafez et al. [10] investigated the mechanical behavior of nano-indentation in a single-crystal FeCoNiCrMn HEA using MD. They found that dislocations produced in the alloy are longer and more concentrated than that in pure Ni. In the deformation process, no amorphization is formed. Fang et al. [11] investigated an HEA composed of face-centered cubic (FCC) and close-packed hexagonal (HCP) phases by MD simulation. They proposed a strong interfacial hardening mechanism for the tensile alloy. The alloy possesses high strength and good ductility. Otto et al. [12] studied the mechanical properties of FeCrNiCoMn HEAs with different grain sizes. They found that the yield strength of the alloy increases with decreasing grain size. Under the deformation process, the effect of fine-grain strengthening diminishes with increasing temperature. Liu et al. [13] studied the deformation mechanism of Al 0.1 CoCrFeNi $\Sigma$3 (111) [11 [combining macron] 0] by MD. They found that under tensile or compressive load, 1/6<112> Shockley partial dislocations are the main slip dislocations during deformation. The grain boundary affects dislocation movement, and dislocation deforms the grain boundary. This deformation mechanism may be the main reason for the strength increasing in an HEA. In Fan et al.'s [14] work, a tensile and scratching HEA was studied by MD. They found that the HEA presents strong anisotropy under uniaxial tensile stress. Shockley dislocation is the transporter of stacking faults deeper into the substrate in the tensile case than that in a scratching one. Ruestes et al. [15] investigated the plastic deformation mechanism of FeNiCrCoCu HEA nano-indentation by MD. They found that the deformation is mainly influenced by dislocation motion. The hardness of the material increases due to the appearance of dislocation interconnections during the deformation process. Nöhring et al. [16] studied the cross-slip of dislocations in NiAl, CuNi, and AlMn FCC solid solution alloys. They showed that the cross-slip is a fundamental process of screw dislocation motion in these alloys. The cross-slip plays an important role in the work hardening and the evolution of the dislocation structure in the alloy. Li et al. [17] investigated a nano-crystalline HEA. They found that with an increase in temperature and a decrease in the strain rate, the grain boundary slip begins to dominate the deformation mechanism. In addition, the occurrence of a phase transition from FCC to body-centered cubic (BCC) can effectively enhance the plasticity of HEAs. Cui et al. [18] investigated the deformation mechanism of an Al-Si single-crystalline material by nano-machining. They found that in nano-machining, the subsurface deformation, dislocation movement, and cutting forces in the material strongly depend on the crystallographic orientation and cutting direction exposed to the material. CrCoNi-based HEAs were studied by Li et al. [19]. They found that the slow diffusion rate of atoms enhances the intermediate temperature performance and mechanical properties of the material. Li et al. [20] studied the structural evolution of an HEA by MD. They found that stacking fault strengthening is induced by two-fold factors. One is from the difference in the stacking fault energies between the HEA matrix and its precipitate phase, and two is the formation of the antiphase domain boundary. The deformation mechanism of an FeCoCrNiCu HEA was studied by Luo et al. [21]. They found that the FeCoCrNiCu HEA undergoes intense dislocation strengthening and significant lattice distortion strengthening during the deformation process. Raturi et al. [22] conducted a mechanistic perspective on the kinetics of plastic deformation in an FCC HEA about the effect of strain, strain rate, and temperature on the mechanistic properties. They suggested that efforts should be directed to develop a theoretical understanding of the dislocation structure and character in a Cantor alloy and employ multi-length and time-scale simulations to understand the activation processes of dislocations.

To the best of our knowledge, studies concerning the effect of the crystallographic direction and temperature on the mechanical properties of HEA are still scarce. In this paper, we refer to the modeling technology and details proposed by Karkalos et al. [23] and use the MD method combined with the dislocation extraction analysis (DXA) technique [24]

to study the effect on the deformation mechanism of an FeNiCrCoMn HEA. It is found that the dislocations produced in the plastic deformation process of the HEA are mainly 1/6<112> Shockley dislocations. The dislocations produced under normal stress loads are far more than that in the shearing process. FCC transformation into HCP does not occur almost until yield stress appears. The yield stress and Young's modulus of the single crystal show strong anisotropy. Young's modulus of the single/double crystal under compressive and tensile loads presents an obvious asymmetry, while there is only a small difference in the polycrystal. Unlike the single crystal, the polycrystal behaves as an isotropic and has strong ductility. The yield stress, yield strain, and Young's modulus reduce gradually with increasing temperature. The strain point is found to be the same for stress yielding, phase transition, and dislocation density, varying from slow to fast with strain at the considered temperature

## 2. Theory and Methodology

In this paper, the mechanical properties of an FeNiCrCoMn HEA were simulated using LAMMPS software [25] (64-bit29Sep2021-MPI, Sandia National Laboratory, Albuquerque, NM, USA). The FeNiCrCoMn HEA FCC structure was built. Firstly, a Ni single crystal was taken as the primary cell with a lattice constant of $a_0 = 0.352$ nm [26]. Then, the primary cell was extended into a super cell along the X-axis, Y-axis, and Z-axis, respectively. The FeNiCrCoMn HEA was finally obtained by random equal ratio substitution of 20% for Ni using Fe, Cr, Co, and Mn elements. In order to make a comparison with the experiment, a polycrystal with 27 grains was also constructed by Atomsk software [27] (beta-0.11.2, Pierre Hirel, Villeneuve d'Ascq, France). The four different kinds of crystals with different crystalline configurations were constructed. The first one was a single crystal with the X-axis along <100>, the Y-axis along <010>, and the Z-axis along <001>; the second one was a single crystal with the X-axis along <11-2>, the Y-axis along <111>, and the Z-axis along <-110>. The third one was constructed with the same crystalline directions as the second one but included a twin boundary in its center along the Y-axis. Due to the periodic condition along the Y-axis, another twin boundary also appeared periodically and was located at bottom of the crystal. The fourth one was constructed with the same crystalline directions as the first one but it was a polycrystal and included 27 grains. For convenience, after that, we called them crystal 1, 2, 3, and 4, respectively. The size of crystal one was $40a_0 \times 40a_0 \times 20a_0$ and the same as that of crystal four. The size of crystal two was the same as crystal three and was $30\sqrt{6}a_0 \times 30\sqrt{3}a_0 \times 20\sqrt{2}a_0$. For an FeNiCrCoMn HEA crystal, a variety of atomic structures might be generated by random substitution. The resulting atomic structure of the HEA was obtained by minimizing the total energy of the alloy. The four kinds of HEA crystals are shown in Figure 1, and the detailed structural parameters of the HEAs are listed in Table 1. Figure 1a is the atomic structure of an HEA crystal for tensile/compressive loading. Figure 1b is a top view of the atomic structure of an HEA crystal for shearing calculation, where top and bottom fixed atomic layers are colored in yellow and blue, respectively, with 1.0 nm thickness. In Figure 1a,b, the blue, yellow, green, purple, and red spheres are for Fe, Ni, Cr, Co, and Mn atoms, respectively, and the load is described by arrows over/below the two crystals. Figure 1c: the model of crystal 1 ignoring the atomic category is an FCC single-crystal structure. Figure 1d: the model of crystal 2 ignoring the atomic category is an FCC single-crystal structure. Figure 1e: the model of crystal 3 ignoring the atomic category is an FCC double-crystal structure, where both the bottom atomic layer with red-colored atoms and the red line on the top surface represent the two twin boundaries. Figure 1f: the model of crystal 4 ignoring the atomic category is an FCC polycrystal structure, where white atoms represent the polycrystalline boundaries.

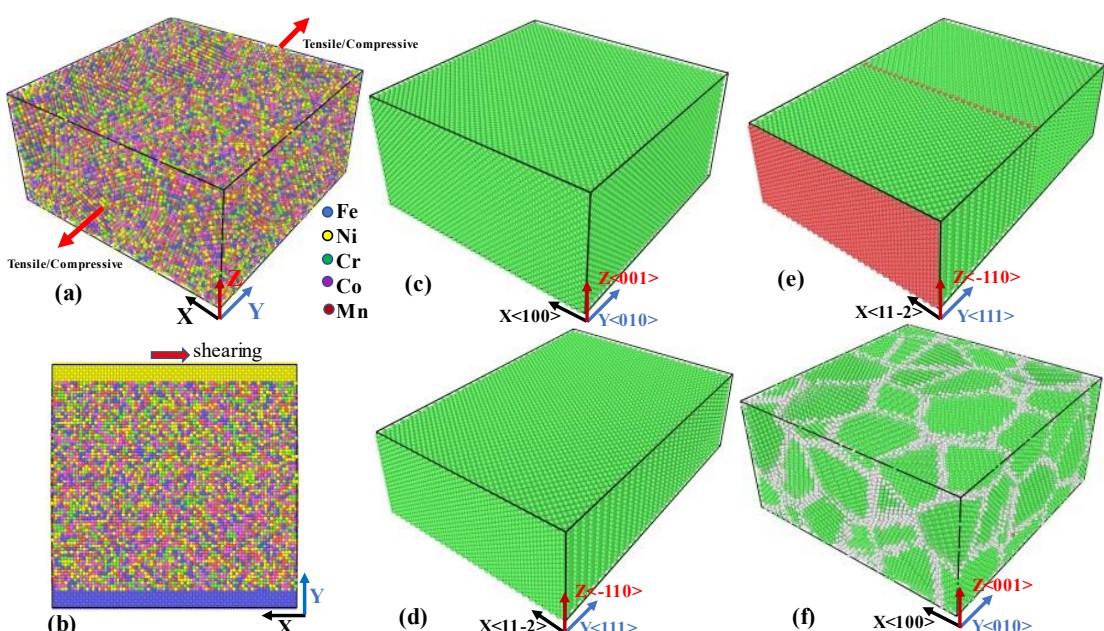

**Figure 1.** (**a**) Atomic structure of an HEA crystal for tensile/compressive loading. (**b**) Top view of atomic structure of an HEA crystal for shearing calculation, where top and bottom fixed atomic layers are colored in yellow and blue, respectively, with 1.0 nm thickness. In (**a**,**b**), the blue, yellow, green, purple, and red spheres are for Fe, Ni, Cr, Co, and Mn atoms, respectively, and the load is described by arrows over/below the two crystals. (**c**) The model of crystal 1 ignoring the atomic category is an FCC single-crystal structure. (**d**) The model of crystal 2 ignoring the atomic category is an FCC single-crystal structure. (**e**) The model of crystal 3 ignoring the atomic category is an FCC double-crystal structure, where both the bottom atomic layer with red-colored atoms and the red line on the top surface represent the two twin boundaries. (**f**) The model of crystal 4 ignoring the atomic category is an FCC polycrystal structure, where white atoms represent polycrystalline boundaries.

**Table 1.** Detailed structural parameters for four crystals of FeNiCrCoMn HEA used for MD simulation, where $a_0$ = 0.352 nm, P expresses the periodic boundary, and S is a shrink-wrapped boundary.

| Crystal Type | Crystal Orientation | Geometric Dimensions | Atomic Number | Compressive(−)/ Tensile(+) Rate | Shearing Velocity | Boundary Condition |
|---|---|---|---|---|---|---|
| Crystal 1 | X<100> Y<010> Z<001> | $40a_0 \times 40a_0 \times 20a_0$ | 128,000 | $\pm 2.0 \times 10^9/s$ | $2.0 \times 10^2 \ m/s$ | Compressive and tensile case: p p p in the X, Y, and Z directions, respectively; shearing case: s s p in the X, Y, and Z directions, respectively. |
| Crystal 2 | X<11-2> Y<111> Z<-110> | $30\sqrt{6}a_0 \times 30\sqrt{3}a_0 \times 20\sqrt{2}a_0$ | 108,000 | The same | The same | |
| Crystal 3 | The same as crystal 2 | The same as crystal 2 | 108,000 | The same | The same | |
| Crystal 4 | The same as crystal 1 | The same as crystal 1 | 128,000 | The same | The same | |

Throughout the simulation, a Nose–Hoover-type thermostat (Nose Hoover, Livermore, CA, USA) was used for temperature control. Detailed structural parameters for four crystals of an FeNiCrCoMn HEA used for MD simulation are shown in Table 1. From the table, it can be seen that crystal 1, 2, 3, and 4 contain 128,000, 108,000, 108,000, and 128,000 atoms, respectively. In compressive and tensile cases, periodic boundary conditions were applied in the X, Y, and Z directions. In the shearing case, a periodic boundary condition was prescribed in the Z direction, and a shrink-wrapped condition was used in the X and Y directions. The compressive and tensile rate was $-2.0 \times 10^9/s$ and $+2.0 \times 10^9/s$, respectively. The shearing velocity of the top fixed atomic layer was $2.0 \times 10^2 \ m/s$. When the energy of a system evolutes and fluctuates around a certain value over time for a longer

period, i.e., the energy varies steadily around a certain value in a small fluctuating range, we believe that the system arrives at a stable state.

In the present simulation, the embedded atomic potential was used, and its parameters are from reference [26] by Choi et al. This potential was successfully used to describe the microstructure and mechanical behavior of an FeNiCrCoMn HEA, such as stress–strain, irradiation damage, nano-indentation, etc. [10,26,28–31].

## 3. Results and Discussions

### 3.1. Static Properties of FeNiCrCoMn High-Entropy Alloy

To compare with previous works [32,33], the static properties, such as the lattice constant, elastic constants, bulk modulus, and shearing modulus of a crystal FeNiCrCoMn HEA, were calculated firstly by using the potential, and the results are listed in Table 2.

**Table 2.** Lattice constant, elastic constants, bulk modulus, and shearing modulus of FeNiCrCoMn HEA.

| Material | $a_0$ (nm) | $C_{11}$ (GPa) | $C_{12}$ (GPa) | $C_{44}$ (GPa) | B(GPa) | G(GPa) |
|---|---|---|---|---|---|---|
| FeNiCrCoMn | 0.359 | 216.48 | 152.57 | 104.96 | 180.26 | 89.78 |
| | 0.3593 [32] | 235 [33] | 160 [33] | 136 [33] | 180 [33] | 84 [32] |

The first row in Table 2 is from the present calculations, and the second row is from references [32,33]. The lattice constant of the NiFeCrCoMn alloy is determined directly from XRD measurements and equals 0.3593 nm at room temperature [32]. The present molecular dynamics calculations predict the lattice constant to be 0.359 nm. Both the theoretical and experimental results are consistent with each other and within the error range of 1%. From Table 2, it can be seen that the calculated elastic constants $C_{11}$, $C_{12}$, $C_{44}$, bulk modulus B, and shearing modulus G for the FeNiCrCoMn HEA are 216.48 GPa, 152.57 GPa, 104.96 GPa, 180.26 GPa, and 89.78 GPa, respectively, which are in good agreement with the EMTO-CPA calculations [33] and the experimental values [32] of 235 GPa, 160 GPa, 136 GPa, 180 GPa, and 84 GPa. We also calculate the XRD patterns of the HEAs at room temperature, as shown in Figure 2. In the figure, the five positions of the peaks can be located at 44, 50.3, 75, 90.7, and 96 degrees, which are in good agreement with the experimentally measured results of 44, 50.2, 75, 90.8, and 96 degrees [26]. These peaks behave as the XRD peaks of an FCC crystal. This indicates that the alloy indeed has an FCC single-phase microstructure in the present calculations. This is in agreement with the experiment.

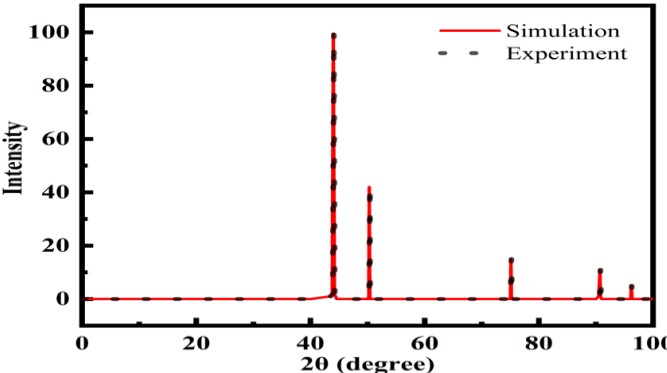

**Figure 2.** XRD patterns of the HEA from the simulation (red line) and experiment (dotted line, see reference [26]).

### 3.2. Compressive/Tensile/Shearing Stress vs. Strain

We calculated the static properties of the HEA, and they are consistent with the experimental or previous theoretical works. Then, we further studied the effect of temperature, crystallographic direction, and twin boundaries on the deformation behavior of the FeNiCrCoMn HEA under uniaxial compression/tension load along the Y-axis and shearing load

along the X-axis. Molecular dynamics simulations are performed at 100, 200, 300, 400, 500, 600, 700, and 800 K, respectively.

The initial configuration is firstly minimized in terms of total energy using a conjugate gradient method to achieve a system with minimal energy at zero temperature. Then, the NPT ensemble is used to simulate the microstructural evolution during the deformation under tensile/compressive load along the Y-direction or shearing load along the X-direction at different simulated temperatures. The time step is set to equal 1 fs, and 30,000 time steps for some temperatures are run to bring the system to an equilibrium state, then followed by 100,000 time steps more at a strain rate of $\pm 2.0 \times 10^9/s$ for the tensile/compressive load and a shearing velocity of $2.0 \times 10^2 \ m/s$ for the shearing process. The strain rate has been used for stress–strain simulations in most materials [34]. In order to attain a quasi-static equilibrium under strain, after each tiny deformation, the system is run for 100 time steps for the equilibrium of the system at a designed temperature for the compressive and tensile cases. In the shearing case, the upper and lower ends with a certain thickness of 1.0 nm along the Y-axis of the simulation box are selected, as shown in Figure 1b. The selected upper and lower ends are set as rigid blocks, where the atoms are fixed. The upper rigid block is displaced along the X-axis at a shearing velocity of $2.0 \times 10^2 \ m/s$, and the lower rigid block is fixed. The atoms in the other region (i.e., except for those in the rigid blocks) make random heat movements by molecular dynamics. DXA [25] and co-neighborhood analysis [35] are used in OVITO software(3.7.11, A. Stuowski, Darmstadt, Germany) [36] to analyze the evolution of dislocations and phase transition under tensile/compressive/shearing cases. The results are shown in Figures 3–9. In each figure, there are twelve panels, where each panel can be identified by the combination of a letter and a number in parenthesis.

The letters a, b, c, and d, in parenthesis, account for crystal 1, 2, 3, and 4, respectively, and the numbers 1, 2, and 3, in parenthesis, express the compressive, tensile, and shearing cases, respectively.

From Figure 3, it can be seen that for the single crystal, the stress linearly increases firstly with increasing strain at various temperatures, and then the maximum stress occurs in each stress–strain curve. This value corresponds to the yield point of stress–strain at the corresponding temperature. This stress is called yield stress, and the corresponding strain is yield strain. When the strain exceeds the yield strain, the compressive/tensile stress in the alloy drops suddenly. In the shearing case, a similar situation can also be seen in the stress–strain curve for crystal one, while for crystals two and three, a slightly different variation in stress with strain is presented. In these two crystals, when strain exceeds the yield strain, the shearing stress in the alloys decreases slowly. In addition, the calculated compressive yield stress in crystal one is in agreement with the experimental values in order of magnitude, while the calculated yield stress of crystals two and three is ten-fold greater than the experimental results, at least. These results indicate that in the process of tensile and compressive load on the FeNiCrCoMn HEA, the single crystal shows strong anisotropy. It should be stressed that large asymmetrical stress occurs in the compressive and tensile process, where the compressive yield stress is far larger than the tensile yield stress in crystals two and three; the situation is just converse in crystal one. The origin of the asymmetry may be quite complex. Two factors may play a key role in the asymmetry. One is that the shape of the potential curve is asymmetrical. This factor may cause the compressive yield stress to be greater than the tensile yield stress when an atom displaces the same distance along the compressive or tensile direction. The other factor is the anisotropy of the single crystals, in which the spacing between the atomic layer exhibits large differences and, at the same time, the pattern of atomic arrangement exhibits a huge variation. The coupled contributions from the two factors may cause an asymmetric effect on the compressive and tensile yield stresses, i.e., when a yield occurs, the deformation arises from the variation in both the atomic configuration and the interaction between atoms; the competition between the two factors determines which one is larger between the compressive and tensile yield stresses. Of course, the strain rate also produces an influence

on the asymmetry. The asymmetrical mechanism should be similar to the ones from the atomic arrangement.

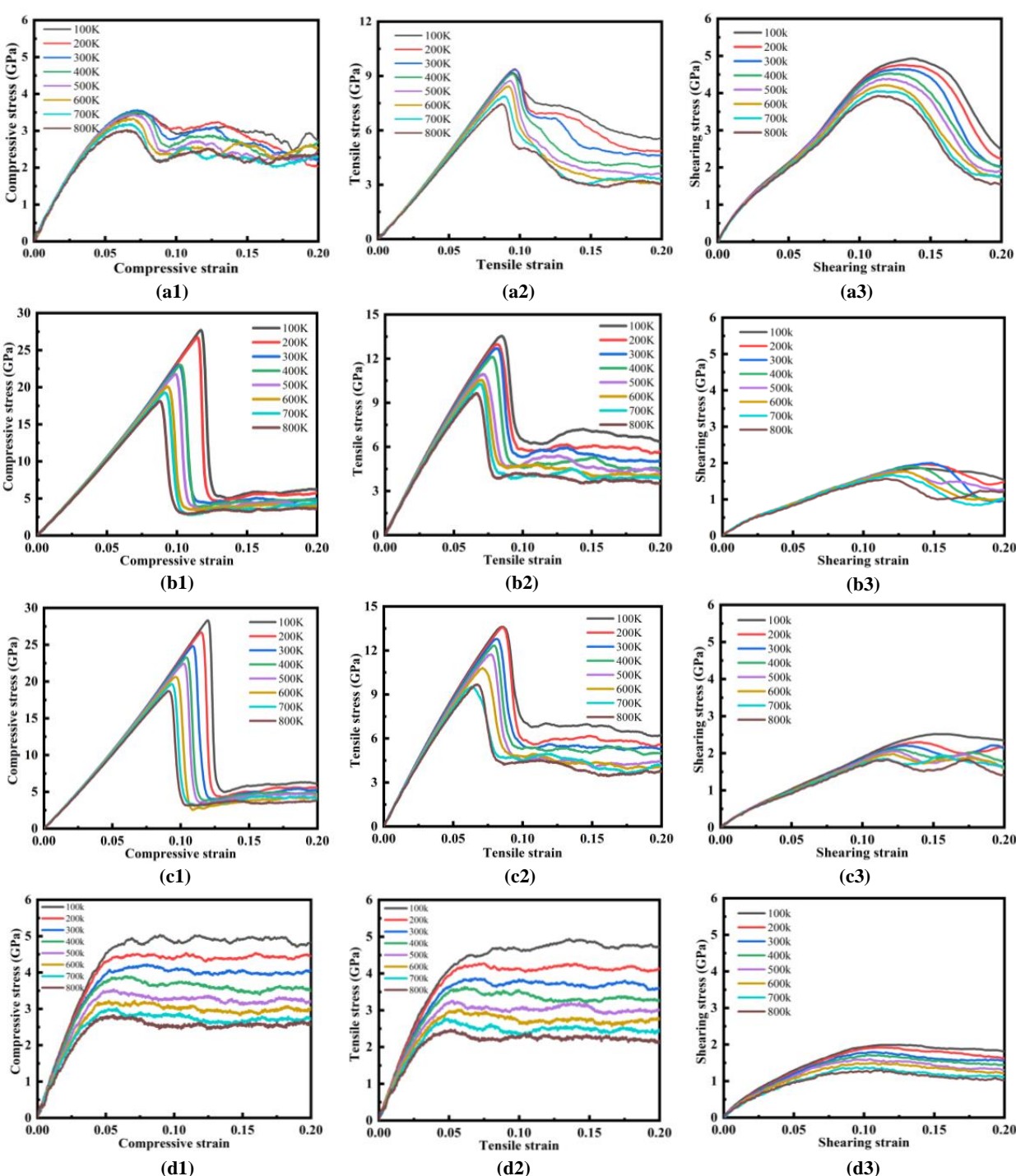

**Figure 3.** Stress vs. strain curve at different temperatures, where (**a1**–**a3**) is for crystal 1 under compressive, tensile, and shearing load, respectively; (**b1**–**b3**) is for crystal 2 under compressive, tensile, and shearing load, respectively; (**c1**–**c3**) is for crystal 3 under compressive, tensile, and shearing load, respectively; (**d1**–**d3**) is for crystal 4 under compressive, tensile, and shearing load, respectively.

From Figure 3d1–d3, it can be seen that for the polycrystal, the stress linearly increases firstly with increasing strain at various temperatures, and then the stress approaches an approximate "saturation" state in each stress–strain curve under tensile and compressive load. In the shearing process, the results are similar to the ones of normal loads, and a

minor difference is that the shearing yield stress decreases slightly with increasing strain after the stress produces a yield. The compressive yield stress is found to be almost identical to the tensile yield stress. All of the characteristics in the stress–strain curve indicate that, unlike the single crystal, the polycrystal behaves as an isotropic and has strong ductility.

From Figure 3, it can also be seen that the variation in stress with strain depends on the temperature and crystalline orientation. For example, under T = 300 K, the compressive/tensile strain at the yield point of crystal one is about 0.0728/0.0962, and the yield stress is about 3.5715/9.351 GPa. Meanwhile, we also considered the polycrystalline case with 27 grains under the same geometric parameters as crystal one, the compressive/tensile strain at the yield point of the crystal is about 0.0802/0.0684, and the corresponding yield stress is about 4.22/3.87 GPa. The experimental compressive/tensile yield stress of the HEA is 1.76/0.7 GPa, and the corresponding strain is 0.219/0.5 [12,37]. The theoretical compressive yield stress is in agreement with the experiment in order of magnitude, while the calculated tensile yield strains (the tensile yield stress) are much lower (higher) than the respective experiment [12,37]. Perhaps, this may arise because the simulated strain rate is $11 \sim 13$ orders of magnitude larger than the experimental strain rate. The strain rate in the current MD simulation is $2.0 \times 10^9 /s$ while the exp. strain rate is usually $10^{-2} \sim 10^{-4}/s$ [26]. The corresponding compressive/tensile yield strain is about 0.1012/0.081 for crystal two and 0.1088/0.0814 for crystal three, and the corresponding compressive/tensile yield stress is about 23.0319/12.7097 GPa for crystal two and 24.8388/12.7943 GPa for crystal three. From these numerical results, it can be seen that under the compressive/tensile load along this <111> direction, the calculated yield stress is far larger than the experimental results. Furthermore, it can also be seen that the yield stress/strain of crystal two is almost identical to the one of crystal three. This indicates that the existence of a single grain boundary only produces an extremely tiny and weak effect on the compressive/tensile yield stress/strain. In other words, it can be concluded that the twin boundary produces only a limited effect on the yield stress/strain, while the crystalline orientation has a significant on the yield stress/strain of the FeNiCrCoMn HEA.

It can also be seen from Figure 3 that under the same temperature in crystal one, the compressive yield stress is numerically smaller than the tensile yield stress. However, in crystals two and three, the situations are opposite to the one in crystal one, i.e., the compressive yield stress is greater than the tensile one in crystals two and three. From Figure 3, it can be also seen that with increasing temperature, the yield stress of the FeNiCrCoMn HEA under compressive/tensile load decreases gradually. This will be shown further in a later section, and the results are consistent with the results from references [33,38]. In the process of uniaxial compression/tension, the yield stress of the polycrystal is comparable to that of crystal one. The stresses of the two crystals decrease with the increase in temperature. The polycrystal has better ductility than that of the single crystal.

Young's modulus of an HEA can be calculated by the slope of the stress–strain curve in the elastic deformation stage. For example, in Figure 3a1,a2, Young's modulus is calculated to be 67.38/99.3 GPa under the compressive/tensile stress of crystal one at a temperature of 300 K, and the result is 62.51/88.89 GPa at a temperature of 800K. Young's modulus of crystal one under compressive/tensile load decreases by 7.3/15.7% at temperatures from 300 K to 800 K. This decreasing amount in Young's modulus in the tensile case is greater than that in the compression case in crystal one. Unlike crystal one, in crystals two and three, the decreasing amount in Young's modulus in the compressive case is slightly greater than that in the tensile case in the same temperature range. In detail, at 300 K, the tensile/compressive condition for Young's modulus of crystal two is calculated to be equal to 173.6/221.5 GPa, and at 800 K, the corresponding value is 157.7/207.5 GPa. At 300 K, the tensile/compressive condition for Young's modulus of crystal three is calculated to be equal to 174.0/221.1 GPa, and at 800 K, the corresponding value is 158.0/207.0 GPa. Meanwhile, at 300 K, the tensile/compressive condition for Young's modulus of crystal four is calculated to be equal to 94.75/99.37 GPa, and at 800 K, the corresponding value is 71.62/75.08 GPa. From these results, it can be seen that Young's modulus of crystal

two is in agreement with that of crystal three. Furthermore, it can be concluded that the twin grain boundary produces little effect on Young's modulus, while the crystallographic direction has a greater effect on the modulus at different temperatures. We also studied the shearing properties of the three single crystals and the polycrystal. The results are shown in Figure 3a3–d3. During the shearing process of the FeNiCrCoMn HEA along the X-axis, the shearing stress–strain curves of the three single crystals at different temperatures are slightly different. The effect of crystalline orientation on the shearing yield stress of the material is relatively obvious, while the effect on strain is weak. For example, at 300 K, the shearing yield strain/stress of crystal one is calculated to be 0.1263/4.64 GPa; crystal two, 0.1449/1.99 GPa; crystal three, 0.1298/2.19 GPa; and the polycrystal, 0.1071/1.79 GPa. In Ref [26], the calculated shearing yield strain/stress of the crystal is 0.08/3.1GPa. This is identical to the present results for shearing yield stress and slightly smaller than the present works for shearing yield strain. For crystals two and three, the present calculated results are slightly smaller than the results of the crystal from Ref [26] and the present results from crystal one. It can be concluded that under shearing load, the twin boundary produces only a limited effect on the shearing yield stress and strain, while the crystalline orientation has a heavy effect on the shearing yield stress/strain of the FeNiCrCoMn HEA.

We calculated the dislocation distributions and the variation in dislocation density with strain. The results are shown in Figures 4 and 5. From Figures 4 and 5, it can be seen that various dislocations are generated during the plastic deformation process. Combined with Figure 3, it can also be found that the dislocations are almost not generated until plastic deformation occurs. There are 1/2<110> perfect dislocation, 1/6<112> Shockley dislocation, 1/6<110> stair-rod dislocation, 1/3<100> Hirth dislocation, 1/3<111> Frank dislocation, etc. As a well-known fact, in an FCC structure, the perfect dislocation may be decomposed into two partial dislocations, and a stacking fault links the two partial dislocations. An FeNiCrCoMn HEA is just an FCC crystal. A 1/2<110> perfect dislocation can easily be decomposed into 1/6<211> and 1/6<12-1> partial dislocations in the (111) plane of an FeNiCrCoMn HEA [37–40]. This is a possible reason that 1/6<112> Shockley dislocations account for a larger proportion than other kinds of dislocations, as shown in Figure 4. Otherwise, a more likely reason may be that the active energy of a 1/6<211> dislocation is the lowest. The existence of a large number of 1/6<112> partial dislocations in the FeNiCrCoMn HEA during plastic deformation also indicates that 1/6<112> Shockley dislocations in the FCC FeNiCrCoMn HEA are more stable than other types of dislocations. Compared with compressive/tensile loading, under shearing load on crystals one, two, three, and four, only a few dislocations are produced. The reason might stem from the fact that during the shearing process, the shearing face is just a sliding face and a generated dislocation easily disappears at the end of the face. Unlike under the compressive/tensile case, from Figure 4a3–d3, it can be found that all the generated dislocations almost belong to the 1/6<112> Shockley dislocations under the shearing case. The shearing faces of crystals two and three are the easy sliding (111) faces. As a well-known fact, a Shockley-type dislocation is easier to be initiate and slide in a (111) face than a (010) face; this leads to the dislocations being fewer in crystals two and three than in crystals one and four. Likewise, a twin boundary is a hurdle to the sliding of dislocation, and this may be the reason that the tensile, compressive, and shearing yield stress of crystal three is slightly greater than the corresponding one of crystal two. In the polycrystal, there are a lot of sliding faces due to the random orientation of many grains; this leads to a lower shearing yield stress than that of a single crystal, in general. From Figure 4, it can also be found that the dislocation line is far shorter in the polycrystal than in the single crystal. The reason stems from that many boundaries "cut down" the dislocation lines in the polycrystal.

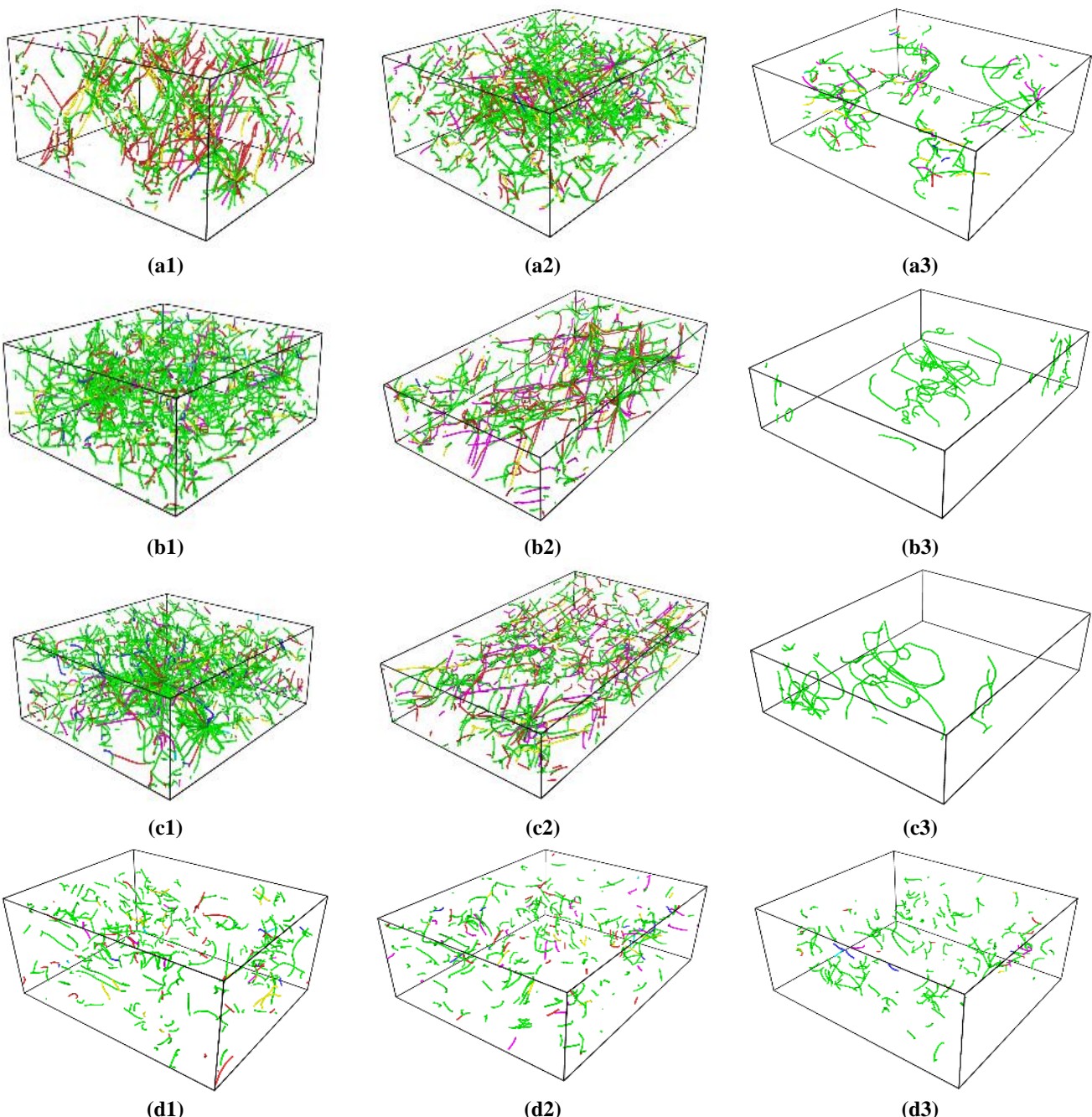

**Figure 4.** The dislocation distribution in HEAs from the snapshots of the compressive/tensile/shearing strain of 0.15 by MD simulations at a temperature of 300 K, where (**a1**–**a3**) is for crystal 1 under compressive, tensile, and shearing load, respectively; (**b1**–**b3**) is for crystal 2 under compressive, tensile, and shearing load, respectively; (**c1**–**c3**) is for crystal 3 under compressive, tensile, and shearing load, respectively; (**d1**–**d3**) is for crystal 4 under compressive, tensile, and shearing load, respectively. In the each subfigure, the blue, green, violet, yellow, cyan, and red lines represent 1/2<110> perfect, 1/6<112> Shockley, 1/6<110> stair-rod, 1/3 <100> Hirth, 1/3 <111> Frank dislocations, and the other types of dislocations, respectively.

Figure 5 shows the variation in the total dislocation density vs. the compressive/tensile/ shearing strain at different temperatures. It can be found that for crystals one, two, and three, there exists a point at which the total dislocation density increases slowly at its incipient plastic deformation stage, and then suddenly increases with the increasing strain. We call this point a transition point. Comparing Figures 3 and 5, we can see that the

yield point of the stress (shown in Figure 3) is just corresponding to the transition point of the curve of total dislocation density vs. strain (see Figure 5). This indicates that when a large number of dislocations occur incipiently, the stress of the HEA starts to yield. Roughly speaking, when such a strain is loaded continuously, the total dislocation density continuously increases until a "maximum value", and then it converges to a stable value approximately, as shown in Figure 5. From Figure 5, it can be seen that at the transition point between the incipiently occurring and rapidly increasing total dislocation concentration, the strain at the yield point decreases approximately with increasing temperature. Note that the total dislocation density at the transition point almost comes from the Shockley-type dislocation of 1/6<112>, as shown in Figure 5. At the convergence stage (strain is larger than about 0.10 for the compressive and tensile load, and about 0.15 for the shearing load), the total dislocation density is the largest at room temperature (see Figure 5). It is worth stressing that the strain yield point in Figure 3 is identical to the transition point, where the variation in total dislocation density vs. strain changes from incipiently slow to consequently rapid in Figure 5. This indicates that only if the dislocations accumulate to a sufficient number can the yield stress in a material take place. It can also be seen from Figure 5 that the total dislocation density decreases with the increase in temperature, and the point corresponding to the rapid increase of dislocation density tends to be a smaller strain with the increase in temperature. This is because that with the increase in temperature, the thermal activation energy of dislocation sliding increases [39]. From Figure 5, it can be seen that the higher the temperature is, the easier the dislocation slip is. In this case, the annihilation between dislocations occurs easily. This may also be the reason why the total dislocation density decreases with increasing temperature under the same strain conditions. It can be seen from Figure 4 that the FeNiCrCoMn HEA may accumulate a large number of dislocations in the plastic deformation stage. Dislocation accumulation leads to stress concentrations (SC). When the strain increases further, the SC makes stress yield. Generally speaking, plastic deformation leads to the generation and movement of dislocations. This movement makes dislocations interact, cross-connect, recombine, and even interact with another phase, such as the HCP phase, if present. All these factors lead to the accumulation of dislocations that do not easily slide, thus, hardening the material [39]. The same phenomenon can also be observed in the polycrystalline HEA with 27 grains. For example, in Figure 3d1, the compressive stress is located at a local minimum value at the 0.075 strain of 100 K. When the strain is further compressed, the stress bounces continuously until the strain of 0.09. This is a work-hardening process in polycrystal deformation. Due to the existence of grain boundaries in polycrystals, there are a certain number of dislocations at the initial stage of deformation. With the continuous loading of strain, the dislocation density increases further at the yield stage and tends to gradually stabilize. The variation in the total dislocation density with strain in the range of the considered temperature is shown in Figure 5.

The variation in the total length of the different types of dislocations with strain is shown in Figure 6. It can be seen that the dislocations produced in the tensile, compressive, and shearing process of FeNiCrCoMn HEA are mainly 1/6<112> Shockley dislocations. The stair-rod dislocation and Hirth dislocation are produced in small numbers. In crystal one, the number of Shockley dislocations is higher in the tensile process than in the compressive case, while an inverse situation is found in crystals two, three, and four. Compared with the shearing load, their amounts are all far more than the ones in the shearing process. Crystal four has a lot of grain boundaries and a certain number of dislocations. Thus, a much smaller number of dislocations are present in the polycrystal than in the single/double crystal, as shown in Figures 5 and 6. This may be attributed to a lot of dislocations disappearing into grain boundaries as a sinking source. The same reason was understood previously, i.e., a generated dislocation in the shearing case disappears easily at the end of the shearing face. The variation in the other types of dislocation lengths with strain is also shown in Figure 6.

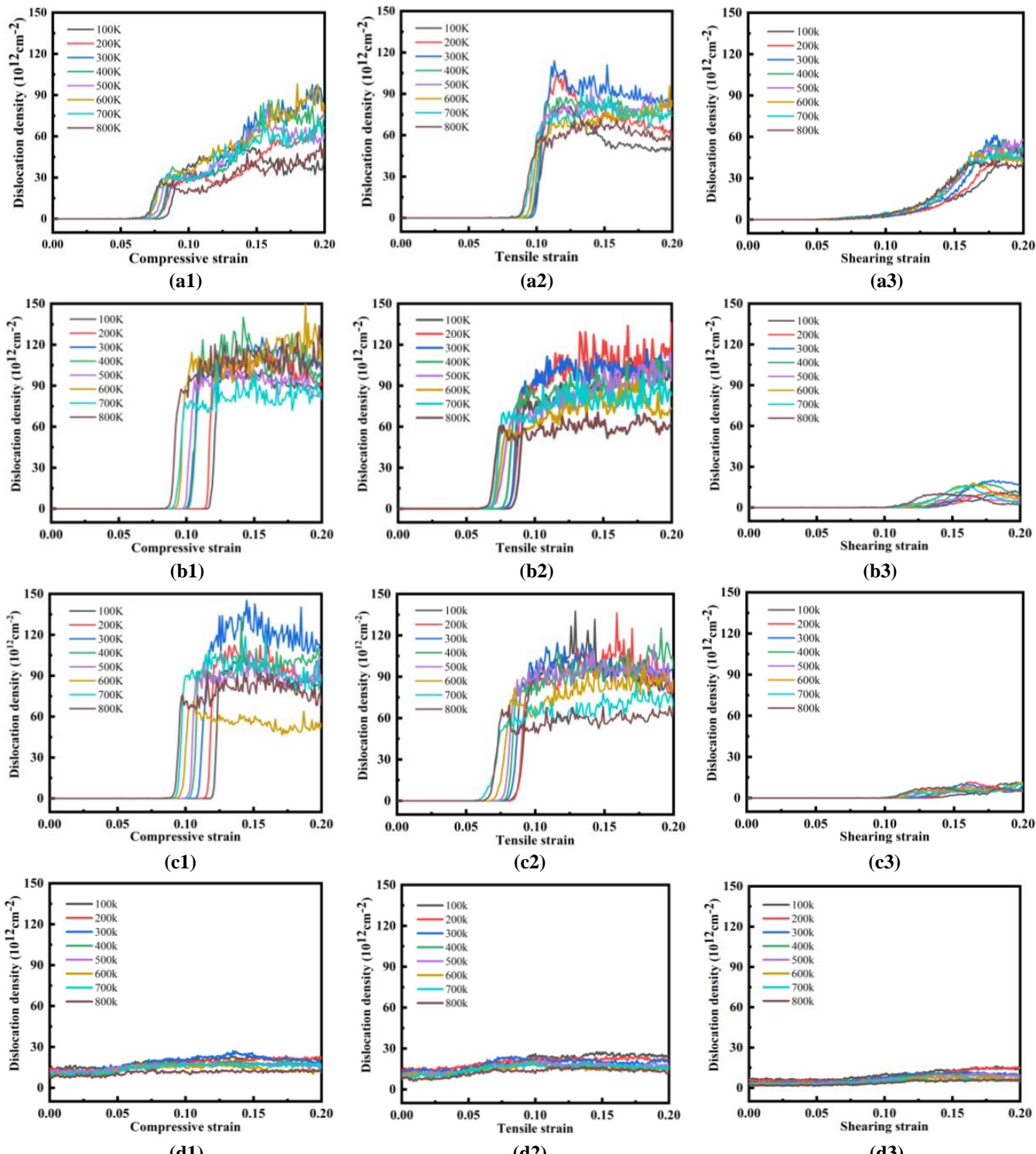

**Figure 5.** Total dislocation density vs. compressive/tensile/shearing strain at different temperatures, where (**a1–a3**) is for crystal 1 under compressive, tensile, and shearing load, respectively; (**b1–b3**) is for crystal 2 under compressive, tensile, and shearing load, respectively; (**c1–c3**) is for crystal 3 under compressive, tensile, and shearing load, respectively; (**d1–d3**) is for crystal 4 under compressive, tensile, and shearing load, respectively.

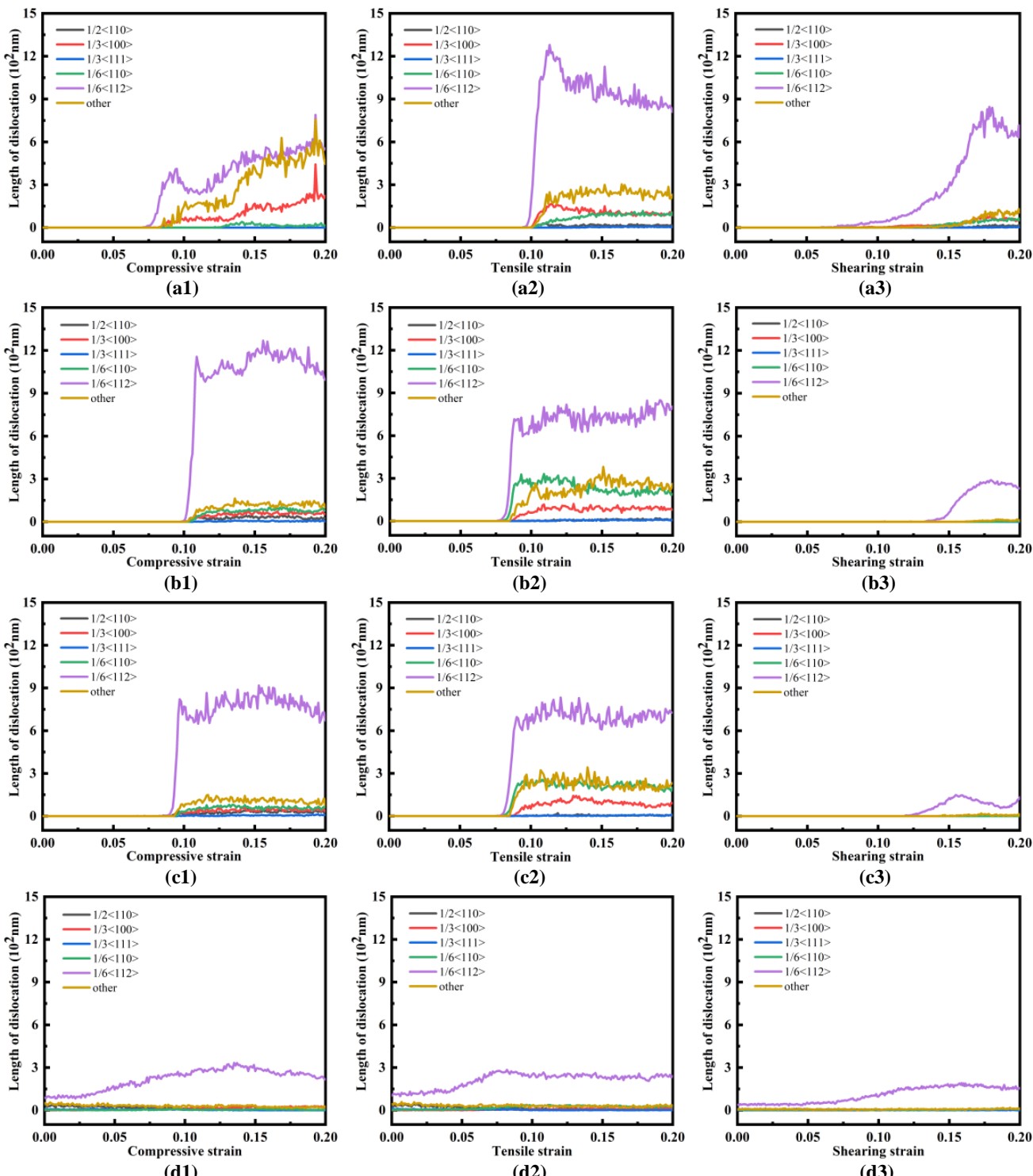

**Figure 6.** The variation in different kinds of dislocation lengths with the compressive/tensile/shearing strain at 300, where (**a1**–**a3**) is for crystal 1 under compressive, tensile, and shearing load, respectively; (**b1**–**b3**) is for crystal 2 under compressive, tensile, and shearing load, respectively; (**c1**–**c3**) is for crystal 3 under compressive, tensile, and shearing load, respectively; (**d1**–**d3**) is for crystal 4 under compressive, tensile, and shearing load, respectively.

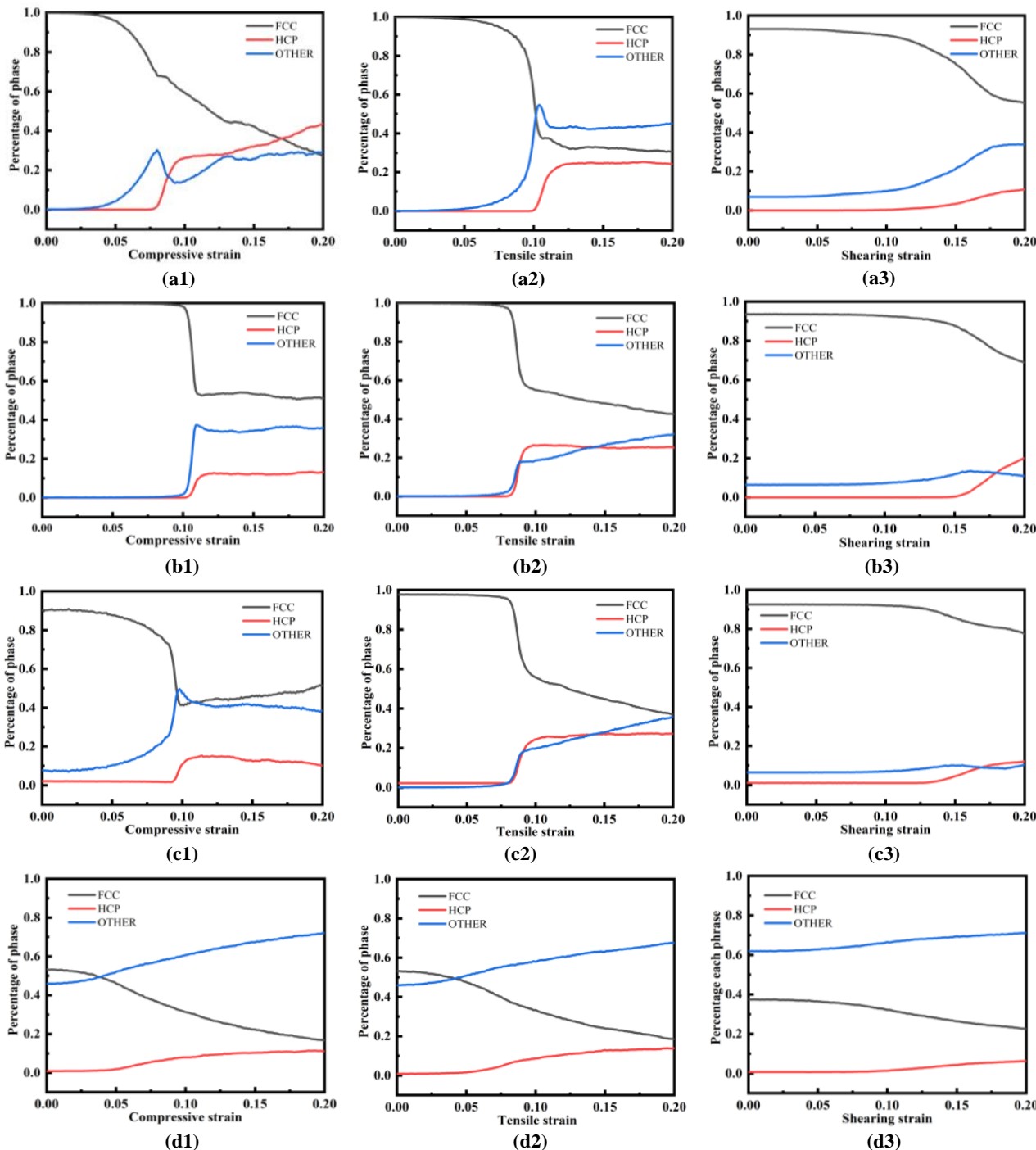

**Figure 7.** Phase concentration as a function of compressive/tensile/shearing strain at 300 K, where (**a1**–**a3**) is for crystal 1 under compressive, tensile, and shearing load, respectively; (**b1**–**b3**) is for crystal 2 under compressive, tensile, and shearing load, respectively; (**c1**–**c3**) is for crystal 3 under compressive, tensile, and shearing load, respectively; and (**d1**–**d3**) is for crystal 4 under compressive, tensile, and shearing load, respectively.

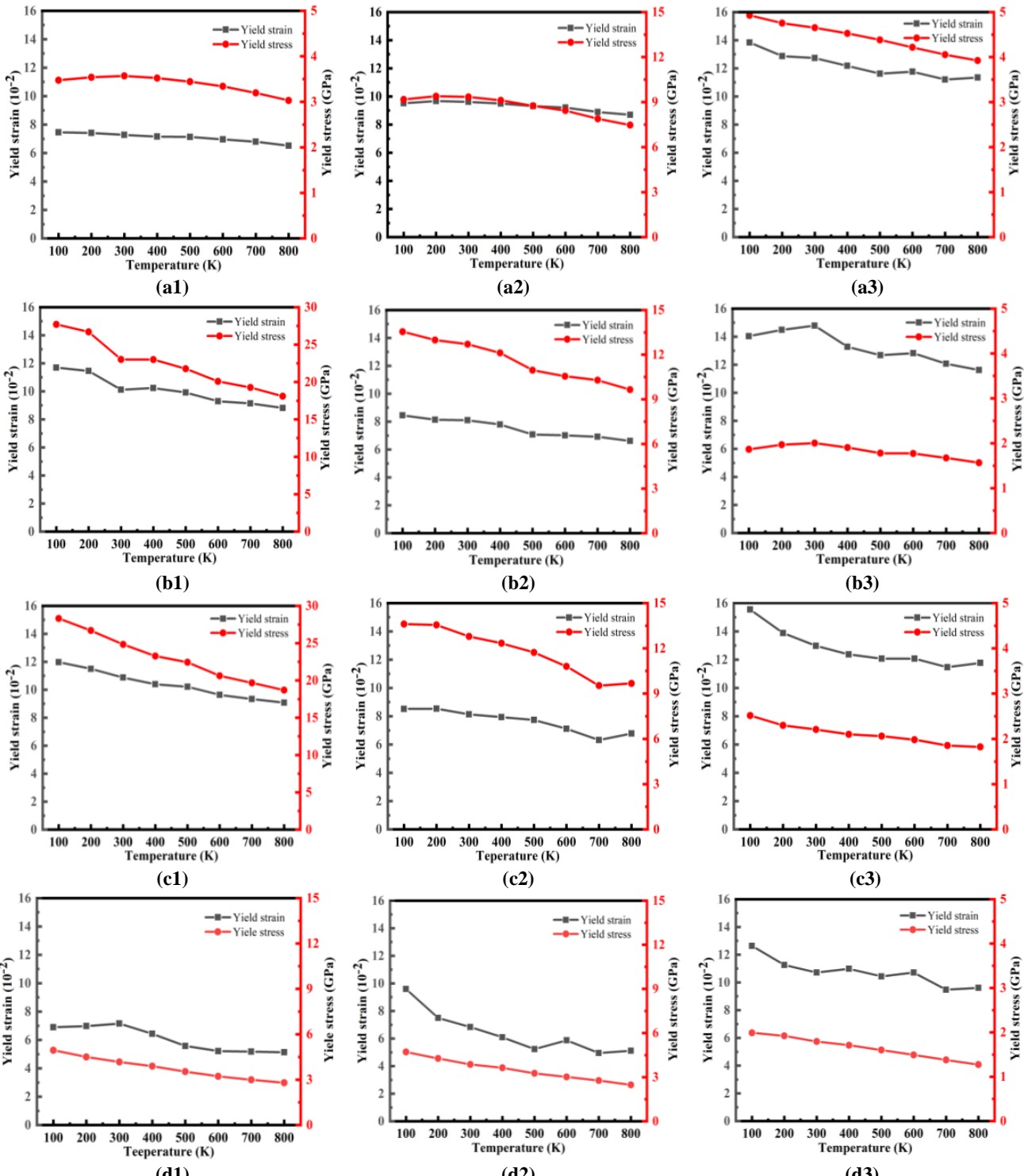

**Figure 8.** Compressive/tensile/shearing yield strain/stress as a function of temperature where (**a1**–**a3**) is for crystal 1 under compressive, tensile, and shearing load, respectively; (**b1**–**b3**) is for crystal 2 under compressive, tensile, and shearing load, respectively; (**c1**–**c3**) is for crystal 3 under compressive, tensile, and shearing load, respectively; and (**d1**–**d3**) is for crystal 4 under compressive, tensile, and shearing load, respectively.

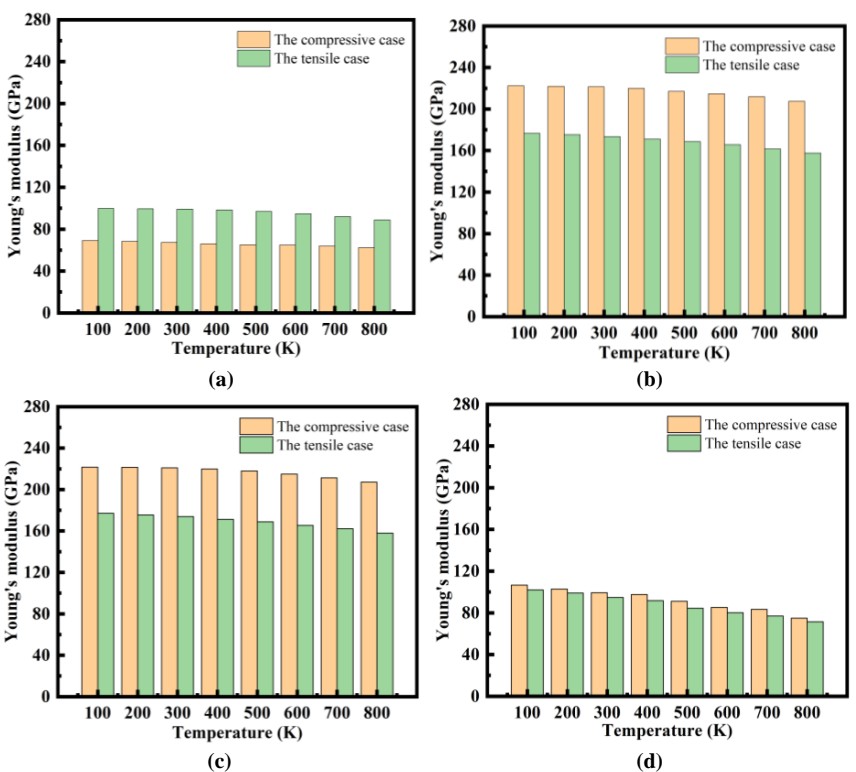

**Figure 9.** Young's modulus of crystals under compressive/tensile load as a function of temperature: (**a**) crystal 1, (**b**) crystal 2, (**c**) crystal 3, and (**d**) crystal 4.

### 3.3. Phase Transition of FCC into HCP and Temperature-Dependent Yield Stress/Strain

Figure 7 depicts the variation in the phase structure in the HEA with strain at 300 K. As shown in the figure, some FCC structures of the HEA begin to transform into HCP and other unknown structures at the yield point. It can be seen from the figure that HCP begins to appear when the strain crosses the yield point of load on the four crystals. At the same time, fracture occurs in crystal one under all kinds of loads. Fracture appears on crystals two and three only under normal load but does not appear under shearing load. Fracture never appears in the polycrystal, although the strain exceeds the yield point under the three kinds of loads (see Figure 3d1–d3). Thus, the polycrystal exhibits far stronger ductility than the single/double crystal. Because the stacking fault energy of the FCC structure (111) is the lowest, a 1/2<110> perfect dislocation is difficult to generate; instead, two 1/6<211> and 1/6<12-1> partial dislocations are generated on the plane. This makes it possible to convert part of the FCC structure into the HCP structure. Similar phenomena can also be observed at other temperatures, which are not shown here. From the figure, it can be seen that the proportion of HCP produced depends on the crystalline orientation. The number of produced HCP phases is greatest in crystal one, and the number of produced HCP phases in crystal two is the same as in crystal three. HCP is fewer in the shearing case than in the tensile/compressive case. Thus, it can be concluded that a single twin boundary has little influence on the generation of the HCP phase. Before the deformation of the polycrystalline HEA, a certain number of non-FCC phases are present due to the existence of grain boundaries. As stress is continuously loaded, the FCC phase begins to transform into the HCP phase and the other unknown phase, and then the HCP and other phase are continuously produced. Under normal load, the other phase is higher in the polycrystal than in the single/double crystal, while in the shearing case, the amount of variation in the FCC phase into both the HCP and other phase with increasing strain in the polycrystal is very similar to that in the single/double crystal (see Figure 7).

As a well-known fact, the hardening of the material mainly stems from the aggregation of dislocations, and this enhances the resistance of the material to deformation. Figure 8

shows the change of yield stress/strain with temperature. It can be seen from the figure that the compressive yield strain is smaller than the tensile yield strain with increasing temperature in crystal one, while the situation is inverse in crystals two and three. It is a similar situation in crystal four to crystal one. In the shearing case, the yield strain with increasing temperature is larger than the one under normal load for the four crystals. It can be also seen that the yield stress decreases with the increase in temperature in the four crystals. The yield strain/stress decreases with increasing temperature, which is consistent with the experimental results [12,37].

The variation in Young's modulus with temperature is shown in Figure 9. Figure 9a–d correspond to crystals one, two, three, and four, respectively. From the figure, it can be seen that Young's modulus decreases with increasing temperature for the four crystals under normal load. It can be understood that the higher the temperature is, the greater the atomic spacing is and the weaker the binding force between atoms is. The lower the temperature, the lower the potential energy of the system and the smaller the atomic vibration. Thus, the lattice is less prone to deformation. In this situation, Young's modulus reduces with increasing temperature. From the figure, it can be also seen that Young's modulus under tensile load is always larger than the one under compressive load in crystal one, while the situation is inverse in crystals two, three, and four. In addition, Young's modulus of crystals two and three is about two times larger than the one of crystals one and four. Under the considered temperature, Young's modulus of the single/double crystal in compressive and tensile loads presents an obvious asymmetry, while there is only a small difference in the polycrystal. From these results, it can also be genuinely understood that the single crystal behaves as a strong anisotropic. The difference in Young's modulus values should stem from the same factors as mentioned in the previous sections, i.e., the shape of the potential curve and the anisotropy of the single crystal.

## 4. Conclusions

The temperature-dependent microstructure and mechanical properties of the FeNiCr-CoMn HEA were simulated by molecular dynamics. The calculated lattice constant, elastic constants, bulk, and shearing modulus were in agreement with the respective experimental/early theoretical values. The calculated theoretical compressive stress of 3.57 GPa along the <010> direction (4.22 GPa for the polycrystal) was consistent with the experimental results of 1.76 GPa in order of magnitudes. The calculated XRD pattern was also in agreement with the experimental results. It was found that with increasing temperature, Young's modulus and the yield stress/strain of the alloy under compressive/tensile/shearing conditions reduced. This tendency was consistent with the experimental results. A large number of dislocations occurred incipiently as the stress of the HEA started to yield. The dislocations produced in the load process of the FeNiCrCoMn HEA were mainly 1/6<112> Shockley dislocations. The stair-rod dislocation and Hirth dislocation were present in very small amounts. It was found that the Shockley-type dislocations of 1/6<112> appeared earlier and earlier with increasing temperature. At the beginning of yield, the dislocation density increased rapidly with increasing strain for the single/double crystal under normal load until fracture, while, except for crystal one, this situation did not occur under the shearing case and also did not occur in the polycrystal under the three loads. In this stage, part of the FCC structure was destroyed gradually and transformed into the HCP structure or other unknown phase. This depended on the crystalline orientation along which the deformation occurs. Even so, the transformation of the FCC structure into the HCP structure did not occur almost until the stresses yielded. Under normal load, the other unknown phase was greater in the polycrystal than in the single/double crystal, while under the shearing case, the amount of variation in the FCC phase into both HCP and the other phase with increasing strain in the polycrystal was very similar to one in the single/double crystal. In crystal one, the number of Shockley dislocations was more in the tensile process than in the compressive case, while an inverse situation was found in crystals two, three, and four. Compared with the shearing load, their amounts were all far

more than the ones in the shearing process. A much smaller number of dislocations were present in the polycrystal than in the single/double crystal. The low stacking fault energy of the (111) face is a possible reason that the 1/6<112> Shockley dislocations accounted for more proportion than the other kinds of dislocations. Unlike the single/double crystal, in the three load processes, the stress of the polycrystal did not decrease with an increase in strain. The compressive yield strain was smaller than the tensile yield strain with increasing temperature in crystal one, while the situation was inverse in crystals two and three. It was a similar situation in crystal four to crystal one. In the shearing case, the yield strain with increasing temperature was larger than the one under normal load for the four crystals. It could also be seen that the yield stress decreased with an increase in temperature in the four crystals. Young's modulus under tensile load was always larger than the one under compressive load in crystal one, while the situation was inverse in crystals two, three, and four. In addition, Young's modulus of crystals two and three was about two times larger than the one of crystals one and four. Under the considered temperature, Young's modulus of the single/double crystal under compressive and tensile loads presents an obvious asymmetry, while there is only a small difference in the polycrystal. In summary, the single crystal presents as an anisotropic, while the polycrystal behaves as an isotropic and has strong ductility.

**Author Contributions:** Conceptualization, J.C. and F.Y.; methodology, J.C. and F.Y.; software, F.Y. and J.C.; validation, J.C., F.Y., Y.Z. and J.L.; formal analysis, F.Y., J.C., Y.Z. and J.L.; investigation, F.Y., J.C., Y.Z. and J.L.; resources, J.C; data curation, F.Y. and C. J.; writing–original draft preparation, F.Y. and J.C.; writing–review and editing, F.Y., J.C., Y.Z. and J.L.; visualization, F.Y.; supervision, J.C.; project administration, J.C.; funding acquisition, J.C. All authors have read and agreed to the published version of the manuscript.

**Funding:** The project was supported by State Key Laboratory for Advanced Metals and Materials, University of Science and Technology Beijing, Beijing, China (grant no. 2022ZD03).

**Data Availability Statement:** No additional data.

**Conflicts of Interest:** The authors declare no conflict of interest.

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
