# Peer review of "Temperature and Crystalline Orientation-Dependent Plastic Deformation of FeNiCrCoMn High-Entropy Alloy by Molecular Dynamics Simulation"

_metals, doi:10.3390/met12122138_

Round 1
Reviewer 1 Report
Good afternoon, respected editor of the journal and colleagues!
I bring to your attention my opinion about the article "Temperature-dependent plastic deformation of FeNiCrCoMn high-entropy alloy by molecular dynamics simulation"
In this paper was calculated static properties of FeNiCrCoMn high-entropy alloy along different directionin in wide temperature range in agreement with the respective experimental and theoretical results.
After reading the article, a number of questions arose:
How do you explain the pronounced asymmetry during compression and tension deformation?
What is the reason for the difference in Young's modulus values?
How does the model used calculate the interaction of dislocations with each other?
Why is there no work hardening on the A2 curves, which takes place in the experimental results?
Author Response
Dear Editor and Reviewers,
Thank you for your reviewing and comments on the manuscript. According to the suggestions Major revisions on the manuscript have been performed. Please see the details as follows:
Question 1:How do you explain the pronounced asymmetry during compression and tension deformation?
Answer 1:The origin of the asymmetry may be quite complex. Two factors may play a key role in the asymmetry. One is that the shape of the potential curve is asymmetrical. This factor may cause that yield compressive stress is greater than yield tensile stress when atom displaces the same distance along compressive or tensile direction. The other factor is the anisotropy of the single-crystals, which the spacing between atomic layer behaves as a large difference and at the same time, the pattern of atomic arrangement exhibits a huge variation. Coupled contribution from the two factors may cause an asymmetric effect on the compressive and tensile yield stress, i.e., when a yield occurs, the deformation gives to arise the variation of both atomic configuration and interaction between atoms, the competition between two factors determine which one is larger between the compressive and tensile yield stress. Of cause, the strain rate also produces influence on the asymmetry. The asymmetrical mechanism should be the similar to the ones from atomic arrangement.
All of these are added manuscript in line 268-280 in blue ink
Question 2:What is the reason for the difference in Young's modulus values?
Answer 2: The variation of Young's modulus with temperature is shown in paper Figure 9. Figure 9 (a, b, c, and d) correspond to crystal one, two, three, and four, respectively. From the figure it can be seen that Young’s modulus decreases with the increasing temperature for four crystals under normal load. It can be understood that the higher the temperature is, the greater the atomic spacing is, and the weaker the binding force between atoms is. The lower the temperature, the lower the potential energy of the system and the smaller the atomic vibration. Thus, the lattice is less prone to deformation. In this situation, Young's modulus reduces with the increasing temperature. From the figure it can be also seen that the Young’s modulus under tensile load is always larger than the one under compressive load in crystal one, while the situation is just inverse in crystal two, three, and four. In addition, Young’s modulus of crystal two and three is about 2 times larger than the one of crystal one and four. From this result, it is also sincerely attained that single crystal behaves as a strong anisotropy. The difference in Young's modulus values should stem from the same factors as mentioned in Q1, i.e., the shape of the potential curve and the anisotropy of the single crystals.
The text has added into the manuscript in line 536-556 in blue ink.
Question 3:How does the model used to calculate the interaction of dislocations with each other?
Answer 3:In general, the interaction between dislocations is described directly by Elastic medium Mechanics. While in molecular dynamics simulation, only the interaction between atoms is calculated by potential function. In this case, the interaction between dislocations can be indirectly obtained from the interaction between atoms. Thus, the movement and mutual entanglement between dislocations may be indirectly exhibited by the interaction between atoms.
Question 4:Why is there no work hardening on the A2 curves, which takes place in the experimental results?
Answer 4: In present simulation for polycrystal the stress-strain curve is also shown a work hardening process. For example, in Fig. 3(d1) the compressive stress is located a local minimum value at the strain of 0.075 of 100K. When the strain is further compressed the stress bounces continuously until the strain of 0.09. This is a work hardening process in polycrystal deformation. This paragraph has been added into the manuscript in line 445-448 in blue ink. The factors causing the process has been included in text of line 436-444.
We correct English grammar and type errors as possible as we can. Enclosed the manuscript revised to you.
Best Regards!
Cai j., Yang f.a., Zhang y., and Lin j.p.
2022.12.8

Reviewer 2 Report
This manuscript presents simulations on the determination of mechanical behavior of FeNiCrCoMn high-entropy alloy under different temperatures and crystal orientations. Although this subject may have some interest, the authors should provide more information about the simulations and perform various modifications, before it can be evaluated for publication.
At first, in the abstract section, the authors should present at least an additional sentence in the beginning in order to describe the subject of the paper and its importance.
The Introduction section is very short for a full length article and insufficient to present the subject of the paper, as well as its importance. It is suggested that it is expanded with a least 2-3 additional paragraphs and at least 10-15 more references should be added and discussed regarding the material which is studied and the methods used. More specifically, the references could be about the MD method and its characteristics, simulation of mechanical properties and simulations of relevant materials, such as: doi.org/10.1063/5.0128135, doi.org/10.1007/s40436-016-0155-4, https://doi.org/10.2174/1573413712666160530122851. Moreover, at the end of the Introduction section, the authors should also present the novel features of the present work in detail.
In section 2 the authors should add much more details about their simulations. For example, the boundary conditions of the model, graphs about the energy minimization procedure which indicate the appropriate convergence of the simulations, the timestep employed at different stages of the simulations, the ensembles used, the number of atoms, total simulation time, etc. are essential to be presented in this section.
Moreover, a Table which summarizes the basic information about the simulations should be added and all the details about the three different crystal models should be presented.
The use of Appendix is not necessary. The authors should present a schematic for each of the three different crystals, including the polycrystal one in Section 2.
Did the authors use the same substrate geometry for tensile, compressive and shearing tests and why?
In the Results and discussions section the figures presenting results from the case with the polycrystalline substrates should be appropriately added to figures 2-7, named as (d1), (d2), (d3), etc.
The fundamental mechanical properties e.g. UTS, yield stress, Young's modulus, which were calculated, should be also depicted in bar charts in respect to temperature for every type of crystal simulated including the polycrystalline substrate as well.
The results of tensile, compression and shearing simulations should be further explained, based on the results regarding dislocations and phase transition.
Author Response
Dear Editor and Reviewers,
Thank you for your reviewing and comments on the manuscript. According to the suggestions Major revisions on the manuscript have been performed. Please see the details as follows:
Question 1:The English language and style of the article need to be revised.
Answer 1:We have made a lot of revisions about the language and style of the article. For example, please see line 13-19, line 30-43, line 47-60, line 116-125, line 190-193, line 198-210, line 213, line 220, line 227-243, line 254-257, line 289-290, line 297-299, line332-329, and so on.
Question 2:At first, in the abstract section, the authors should present at least an additional sentence at the beginning in order to describe the subject of the paper and its importance.
Answer 2:We have added sentences to the abstract that describe the topic of the paper and its importance.
“The effect of crystallographic direction and temperature on the mechanical properties of FeNiCrCoMn high-entropy alloy (HEA) is explored by molecular dynamics simulations.”
The other sentences are also added into the abstract. In addition, the title of the manuscript has been modified into “Temperature and crystalline orientation-dependent plastic de-formation of FeNiCrCoMn high-entropy alloy by molecular dynamics simulation ”
Question 3:The Introduction section is very short for a full-length article and insufficient to present the subject of the paper, as well as its importance. It is suggested that it is expanded with a least 2-3 additional paragraphs and at least 10-15 more references should be added and discussed regarding the material which is studied and the methods used. More specifically, the references could be about the MD method and its characteristics, simulation of mechanical properties and simulations of relevant materials, such as doi.org/10.1063/5.0128135, doi.org/10.1007/s40436-016-0155-4, https://doi.org/10.2174/1573413712666160530122851. Moreover, at the end of the Introduction section, the authors should also present the novel features of the present work in detail.
Answer 3:We have added 2-3 additional paragraphs and 9 references suggested about MD method and its characteristics and 1 reference about experiment into the Introduction. We also present the novel features of the present work in more detail. (see sentences in blue ink in Introduction)
For example: “. It is found that the dislocations produced in the plastic deformation process of the HEA are mainly 1/6<112> Shockley dislocations. The dislocations produced in normal stress load are far more than that in the shearing process. FCC transformation into HCP does not occur almost until yield stress appears . The yield stress and Young’s modulus of the single-crystal show a strong anisotropy. The Young’s modulus of single/double-crystal in compressive and tensile loads presents an obvious asymmetry, while only small difference in polycrystal. Unlike the single-crystal, the polycrystal behaves as isotropic and has a strong ductility. The yield stress, yield strain, and Young’s modulus reduce gradually with increasing temperature. The strain point is found to be the same for stress yielding, phase transition, and dislocation density varying from slow to fast with strain at considered temperature.”, and so on. These sentences are added into the Introduction.
Question 4:In section 2 the authors should add much more details about their simulations. For example, the boundary conditions of the model, graphs about the energy minimization procedure which indicate the appropriate convergence of the simulations, the timestep employed at different stages of the simulations, the ensembles used, the number of atoms, total simulation time, etc. are essential to be presented in this section.
Answer 4:We add more details about the simulation about the boundary conditions of the model, graphs about the energy minimization procedure, the timestep employed at different stages of the simulations, the ensembles used, the number of atoms, total simulation time, etc.
For example: When the energy of a system evolutes and fluctuates around some certain value with time for a longer period, i.e., the energy varies steadily around a certain value in a small fluctuating range, we believe that the system arrives at a stable state, and so on, together with Table 1 below
Table 1. Detailed structural parameters for four crystals of FeNiCrCoMn HEA alloy used for MD simulation, where a0=0.352 nm, P expresses periodic boundary and S does a shrink-wrapped boundary.
|
Crystal type |
Crystal |
Geometric |
Atomic number |
Compressive(-)/tensile(+) rate |
Shearing |
Boundary |
|
Crystal 1 |
X<100> Y<010> Z<001> |
|
128000 |
|
compressive and tensile case: p p p in the X, Y, and Z directions, respectively; shearing case: s s p in the X, Y, and Z directions, respectively. |
|
|
Crystal 2 |
X<11-2> Y<111> Z<-110> |
108000 |
the same |
the same |
||
|
Crystal 3 |
the same as Crystal 2 |
the same as Crystal 2 |
108000 |
the same |
the same |
|
|
Crystal 4 |
the same as Crystal 1 |
the same as Crystal 1 |
128000 |
the same |
the same |
All of the these have been added into Section 2 (the adding text in blue ink.) . The number of atoms and boundary conditions and size are described in line 165-175
Question 5:Moreover, a Table which summarizes the basic information about the simulations should be added and all the details about the three different crystal models should be presented.
Answer 5:We have added the table to summarize the basic simulation information, as shown in Answer 4.
Question 6:The use of the Appendix is not necessary. The authors should present a schematic for each of the three different crystals, including the polycrystal one in Section 2.
Answer 6:We added diagrams of six simulating box shown below, (a) and (b) describe HEA structure with atomic types and stress indicators, (c-d) show crystallographic direction and crystal structure ignoring atomic category . the details, please see line 120-147.
Question 7:Did the authors use the same substrate geometry for tensile, compressive and shearing tests and why?
Answer 7: We simulate four crystals, Crystal one and crystal four possess the same size, while crystal one is a single crystal and crystal four is a polycrystal. Crystal two and crystal three possess the same size, while crystal two is a single crystal and crystal three is a double-crystal
Because the crystallographic orientations of crystal one/four is different from one of crystal two/three, the size of crystal one/four is different from one of crystal two/three. The geometries have been shown iin the figure above and explained in line 120-146 of the manuscript. We do so, we want to study the effect of crystallographic direction and temperature on the mechanical properties of FeNiCrCoMn high-entropy alloy.
Question 8:In the Results and Discussion section the figures presenting results from the case with the polycrystalline substrates should be appropriately added to figures 2-7, named (d1), (d2), (d3), etc.
Answer 8:Yes, We have done so in Figures 3-9. The corresponding results and discussions , are presented in line 281-288 for stress-strain of polycrystal, line 398-402 for dislocation-strain, line 507-510 for phase-strain, line 547-556 for Young’s modulus-temperature in blue ink, and et al.
Question 9:The fundamental mechanical properties e.g. UTS, yield stress, and Young's modulus, which were calculated, should be also depicted in bar charts with respect to temperature for every type of crystal simulated including the polycrystalline substrate as well.
Answer 9: The calculation for UST is ambiguous. If do so, in present work, from Fig. 3 (b2, b2,c1,c2) it is obtained that UST of crystal two and three is 27.5/27.0GPa and 28.0/27.5GPa, respectively, under compressive/tensile load in the temperature of 100K. They are almost equal to their respective yield stress. However, from Fig.3(a1, a2, b3, c3, d) UST cannot be determined at all. In this situation we do not present the results about UST, and only Young’s modulus Vs. temperature are exhibited in Fig. 9 and corresponding results are also discussed in line 536-556. Yield stress/strain Vs. temperature have already been depicted in Fig. 8.
Question 10:The results of tensile, compression and shearing simulations should be further explained, based on the results regarding dislocations and phase transition.
Answer 10: We have further explained tensile/compressive/shearing process based on dislocations in line 393, and line 398-402, line 462-464, line 467-472, based on phase transition in line 484-490, line 507-510. The variation of Young’s modulus Vs. temperature is explained in line 536-556 for single/double/ploy-crystal. They are shown in blue ink in the manuscript.
We correct English grammar and type errors as possible as we can. Enclosed the manuscript revised to you.
Best Regards!
Cai j., Yang f.a., Zhang y., and Lin j.p.
2022.12.8

Round 2
Reviewer 2 Report
The authors performed most of the requested modifications to their manuscript. Thus, it can now be recommended for publication.